# DIFFERENTIAL PRIVACY FOR TRANSFORMER EMBEDDINGS WITH NONPARAMETRIC VARIATIONAL INFORMATION BOTTLENECK

## ABSTRACT

We propose a privacy-preserving method for sharing text data by sharing noisy versions of their transformer embeddings. It has been shown that hidden representations learned by deep models can encode sensitive information from the input, making it possible for adversaries to recover the input data with considerable accuracy. This problem is exacerbated in transformer embeddings because they consist of multiple vectors, one per token. To mitigate this risk, we propose Nonparametric Variational Differential Privacy (NVDP), which ensures both useful data sharing and strong privacy protection. We take a differential privacy approach, integrating a Nonparametric Variational Information Bottleneck (NVIB) layer into the transformer architecture to inject noise into its multi-vector embeddings and thereby hide information, and measuring privacy protection with Rényi divergence and its corresponding Bayesian Differential Privacy (BDP) guarantee. Training the NVIB layer calibrates the noise level according to utility. We test NVDP on the GLUE benchmark and show that varying the noise level gives us a useful tradeoff between privacy and accuracy. With lower noise levels, our model maintains high accuracy while offering strong privacy guarantees, effectively balancing privacy and utility.

## 1 INTRODUCTION

Deep learning methods are highly dependent on the availability of data, but sharing data is often limited by privacy concerns. This is especially true for text data, where private attributes such as gender or age are mixed with useful information and can be detected by attackers (Li et al., 2018). Even when sharing the embeddings of text instead, these representations can still contain sensitive information, and an adversary could use techniques like a GAN (Generative Adversarial Network) attack (Hitaj et al., 2017) to reverse-engineer the original input, potentially reconstructing the text and exposing private information. This is especially true for state-of-the-art attention-based models like transformers, where a text embedding consists of many vectors, one per text token. We want to be able to share such transformer embeddings while still addressing privacy concerns.

Differential Privacy (Dwork, 2006) is widely recognized as the benchmark for rigorously quantifying and mitigating privacy risks associated with processing sensitive data. However, integrating differential privacy into machine learning (Abadi et al., 2016; Bassily et al., 2014) remains a persistent challenge, primarily due to the substantial drop in model performance compared to non-private versions.

We take the approach of applying differential privacy at the stage of sharing the data, before applying machine learning. This approach has the advantage that the shared data can be reused for multiple purposes and to train multiple models. However, differential privacy relies on noise to remove information, and many types of data, in particular discrete data such as text, do not have a straightforward model of noise. If we first embed the data with a transformer encoder and then share noisy embeddings, then we can apply this approach to any data which can be embedded with a transformer. The transformer architecture has gained prominence for its flexibility and effectiveness across diverse tasks, in particular text.

In this paper, we propose a method for adding noise to transformer embeddings which results in both retaining useful information and differential privacy guarantees. The key to this method is using a non-parametric variational information bottleneck (NVIB) regularizer to train a noise model which is calebrated to the downstream task. Our proposed nonparametric variational differential privacy (NVDP) model first uses NVIB to train a distribution over transformer embeddings, and then samples from this distribution to get a noisy embedding which can be shared. Using Rényi Divergence as a measure of privacy (Geumlek et al., 2017), we show that NVDP provides an effective tradeoff between level of privacy and usefulness in the downstream task.

This paper makes the following contributions:

- Propose the NVDP model, which uses a NVIB regulariser for attention-based representations to provide differential privacy.
- Show empirically that NVDP provides a useful tradeoff between privacy and utility.
- Show empirically that NVIB regularisation is more effective than VIB regularisation used in an analogous way.

## 2 BACKGROUND

Our proposed method builds on previous work on Rényi differential privacy and Nonparametric Variational Information Bottleneck.

### 2.1 DIFFERENTIAL PRIVACY

Differential privacy has emerged as the leading standard for rigorously addressing privacy leakage during the processing of sensitive datasets (Dwork et al., 2014). It prevents attackers from deducing too much information about the input data solely from the algorithm's outputs.[1] There are two primary settings for differential privacy: global differential privacy (GDP) and local differential privacy (LDP). LDP is designed for scenarios where data comes from end users who do not trust a central data collector or third parties with their raw data (Dwork et al., 2014). Under LDP, each user independently perturbs their data before sharing it, ensuring that sensitive information remains protected even if the data collector is compromised. By introducing noise or randomness at the user level, LDP enables the collection of aggregated insights from a dataset without revealing individual-level details, thus maintaining privacy while still allowing for statistical analysis (Dwork et al., 2014; Abadi et al., 2016). In our work, we aim to protect embeddings of text by sharing noisy embeddings, and thus we apply local differential privacy. To formalize this, differential privacy ensures that for every pair of adjacent inputs $x$ and $x'$ and a randomized algorithm $M$, the distribution of $M(x)$ and $M(x')$ are close to each other. Originally, the distributions of $M(x)$ and $M(x')$ are considered adjacent if they differ by all the attributes of a single record, but in practice the notion of adjacency can vary. For example, Sun et al. (2019) define two sentences as adjacent if they differ by at most 5 consecutive words, where each sentence is split into segments of 5 words.

**Rényi Differential Privacy (RDP)** is a variation of standard $(\epsilon, \delta)$-differential privacy that employs Rényi divergence as a metric to measure the distance between the output distributions $Q$ of $M(x)$ and $Q'$ of $M(x')$ (Geumlek et al., 2017). This approach is especially advantageous for training machine learning models that require differential privacy.

**Definition 2.1.** Given two probability distributions $\mathcal{Q}$ and $\mathcal{Q}'$ defined over domain $\mathcal{Z}$, the Rényi divergence of order $\lambda > 1$ is:

$$D_\lambda(Q||Q') = \frac{1}{\lambda - 1} \log \left( \int_z Q(z) \left( \frac{Q(z)}{Q'(z)} \right)^{\lambda - 1} \right) \tag{1}$$

Using this divergence, we can formally define the privacy guarantee for a mechanism. In our work, we focus on the local privacy setting where the inputs are individual data points.

---

[1]There is no distinction between sensitive information and non-sensitive information, since we can't tell with certainty what an attacker can deduce from the available information. Thus differential privacy simply tries to reduce all information.

**Definition 2.2** (($\lambda, \epsilon$)-Rényi Differential Privacy)**.** A randomized mechanism $M : \mathcal{X} \rightarrow \mathcal{Z}$ satisfies ($\lambda, \epsilon$)-Local Rényi Differential Privacy if for any pair of adjacent inputs $\boldsymbol{x}, \boldsymbol{x}' \in \mathcal{X}$, the following holds:

$$D_\lambda(M(\boldsymbol{x})||M(\boldsymbol{x}')) \leq \epsilon \tag{2}$$

The privacy parameter $\epsilon$, often called privacy budget, measures the privacy loss associated by the output of the algorithm: $\epsilon = 0$ provides perfect privacy, meaning the output is completely independent of the input. As $\epsilon \rightarrow \inf$, the privacy guarantees diminish, offering no protection for the input data. The other parameter $\lambda$ controls the importance of the worst-case privacy violations, reducing to KL-divergence at its lowest value and the maximum log-difference at its highest value.

**Bayesian Differential Privacy (BDP)** offers an alternative interpretation of privacy by focusing on the change in an adversary's belief. As introduced by Triastcyn & Faltings (2020), instead of comparing two adjacent inputs $\boldsymbol{x}$ and $\boldsymbol{x}'$, BDP analyzes how an output from a mechanism $M(\boldsymbol{x})$ changes an adversary's posterior belief about the input $\boldsymbol{x}$ compared to their prior belief. This prior is typically formed from knowledge of the underlying data distribution, which we can denote as $\mathcal{X}$. Formally, this is captured in the ($\epsilon_\mu, \delta_\mu$)-BDP definition:

**Definition 2.3.** Let $M : \mathcal{X} \rightarrow \mathcal{Z}$ be a randomized mechanism applied to individual data points. Then $M$ satisfies ($\epsilon_\mu, \delta_\mu$)-Local Bayesian Differential Privacy if, for all $\boldsymbol{x}$, a data point $\boldsymbol{x}'$ drawn from the data distribution $\boldsymbol{x}' \sim \mathcal{X}$, and all measurable subsets $S \subseteq \mathcal{Z}$, the following holds:

$$\Pr[M(\boldsymbol{x}) \in S] \leq e^{\epsilon_\mu} \Pr[M(\boldsymbol{x}') \in S] + \delta_\mu \tag{3}$$

Here, $\Pr[M(\boldsymbol{x}') \in S]$ marginalizes over both the outputs in $S$ and the datapoints $\boldsymbol{x}' \sim \mathcal{X}$, and $\delta_\mu$ accounts not only for the probability of catastrophic privacy loss from the mechanism itself but also incorporates the uncertainty an analyst has about the true data distribution. This framework provides a practical way to reason about privacy for typical data points, which is highly relevant for machine learning models trained on specific data domains.

**Combining BDP and RDP** Triastcyn & Faltings (2020) also point out that Rényi divergence can be used to define an upper bound on the worst-case privacy loss over sampled outputs. They use this fact to define a privacy accountant which allows multiple accesses to the data, but it can also be used to simply define a new privacy measure which is less sensitive to worst-case loss and closer to expected-case loss over outputs, depending on the setting of $\lambda$. This gives us a privacy measure which summarises the privacy risk over both the distribution of alternative examples $x'$ and the possible outputs. In our experiments, we leverage this connection by first calculating the Rényi Divergence (RD) and then converting it into an interpretable ($\epsilon_\mu, \delta_\mu$)-BDP guarantee. We use the privacy accounting mechanism detailed in Theorem 2 of Triastcyn & Faltings (2020). For the full derivation and implementation details of the accountant, we refer the reader to the original work by Triastcyn & Faltings (2020).

## 2.2 Nonparametric Variational Information Bottleneck

NVIB (Henderson & Fehr, 2023) is a variational information bottleneck regulariser for attention layers. It replaces the set of key vectors in a transformer's attention layer with a latent mixture of impulse distributions, specified as a set of vectors $\boldsymbol{Z}$ and their normalised weights $\boldsymbol{\pi}$. To access this mixture distribution, it generalises the attention function to "denoising attention". It then uses Bayesian nonparametrics to define prior and posterior distributions over these mixture distributions, and a KL divergence with the prior to regularise the amount of information in its posterior.

**Prior** Since NVIB uses Bayesian inference to specify its posterior, it needs a prior over the latent space. Since the number of keys in an attention layer grows with the length of the input and the input can be arbitrarily large, this prior needs to specify a distribution over sets of weighted vectors ($\boldsymbol{\pi} \in \mathbb{R}^m, \boldsymbol{Z} \in \mathbb{R}^{m \times d}$) which have no finite limit to their size $m$. Nevertheless, we can still define probability distributions over this infinite set by applying Bayesian nonparametric methods. In particular, Henderson & Fehr (2023) use a Dirichlet Process (DP) to define the prior distribution $DP(G_0^p, \alpha_0^p)$ over unboundedly large mixtures ("p" designates the prior and "q" designates the posterior). The vectors $\boldsymbol{Z}_i$ are generated by the base distribution $G_0^p$, which is a Guassain distribution with parameters ($\boldsymbol{\mu^p} = \boldsymbol{0}, (\boldsymbol{\sigma^p})^2 = \boldsymbol{1}$). The weights $\boldsymbol{\pi}$ are generated with a symmetric Dirichlet

distribution parameterised by the total pseudo-count $\alpha_0^p = 1$, which tends to generate large weights on a few vectors and a long tail of exponentially smaller weights.

**Posterior** An NVIB layer uses Bayesian inference to parameterise its posterior distribution $DP(G_0^q, \alpha_0^q)$ in terms of a set of pseudo-observations computed from the set of vectors which the transformer inputs to the layer. These $n$ input vectors $\boldsymbol{x} \in \mathbb{R}^{n \times d}$ are each individually projected to a pseudo-count $\alpha_i^q \in \mathbb{R}$ and a pair of Gaussian parameters $(\boldsymbol{\mu_i^q} \in \mathbb{R}^d, (\boldsymbol{\sigma_i^q})^2 \in \mathbb{R}^d)$. The base distribution $G_0^q$ is a mixture of Gaussians, where each pair of Gaussian parameters specifies a component of the mixture, and the pseudo-counts specify their weights. In addition, there is an $n+1^{\text{th}}$ component of the base distribution specified by the prior's parameters, so $(\alpha_{n+1}^q, \boldsymbol{\mu_{n+1}^q}, (\boldsymbol{\sigma_{n+1}^q})^2) = (\alpha_0^p, \boldsymbol{\mu^p}, (\boldsymbol{\sigma^p})^2)$.

$$F \sim \text{DP}\left(G_0^q,\ \alpha_0^q\right) \qquad \alpha_0^q = \sum_{i=1}^{n+1} \alpha_i^q \qquad G_0^q = \sum_{i=1}^{n+1} \frac{\alpha_i^q}{\alpha_0^q} G_i^q \qquad G_i^q = \mathcal{N}(\boldsymbol{\mu_i^q}, \boldsymbol{I}(\boldsymbol{\sigma_i^q})^2) \qquad (4)$$

As in previous work (Fehr & Henderson, 2024), during training our NVIB layer approximates samples of weighted vectors from this posterior as $\boldsymbol{\pi} \sim \text{Dir}(\boldsymbol{\alpha^q})$ and $\boldsymbol{Z_i} \sim \mathcal{N}(\boldsymbol{\mu_i^q}, (\boldsymbol{\sigma_i^q})^2)$, where $Dir(\cdot)$ is a Dirichlet distribution. An NVIB layer also has the capability to drop specific embeddings by setting their pseudo-counts to zero, effectively reducing the complexity of the representation.

**NVIB training objective** The NVIB loss regularises the flow of information through the latent representation. The loss consists of three components: a task loss ($L_T$) and two Kullback-Leibler (KL) divergence terms, $L_D$ and $L_G$. The $L_T$ term serves as a supervised learning objective, guiding the latent representation to retain enough information to perform the task. The $L_G$ term encourages noise in the Gaussian components, making them each less informative. The $L_D$ term both encourages noise in the weights and encourages some of the Dirichlet distribution's pseudo-counts to reach zero, effectively eliminating some vectors and reducing the overall information capacity of the latent representation. Two hyperparameters are added to adjust the impact of both the $L_D$ and $L_G$ terms:

$$L = L_T + \lambda_D L_D + \lambda_G L_G \qquad (5)$$

# 3 NVDP: Nonparametric Variational Differential Privacy

Our goal is to share transformer embeddings in a privacy-preserving manner, but still share embeddings which are useful. Intuitively, RDP measures the degree of privacy by measuring how much information a sampled embedding is expected to convey (with more weight put on the worst case samples). If we know what type of task we intend the embeddings to be used for, then we can measure the degree of utility of an embedding by measuring the accuracy of a model trained on such a task. Reducing the total amount of information in a latent representation while retaining enough information to perform a task is precisely the objective of a variational information bottleneck regulariser. It trains a posterior distribution over embeddings, a sample from this posterior provides limited information, and this information is optimised to perform the downstream task. We apply this general idea to the case of transformer embeddings, using NVIB as the regulariser and RDP as the privacy measure. We propose nonparametric variational differential privacy (NVDP) as a method to learn posterior distributions over embeddings which provide good RDP privacy and are calibrated to a given utility objective and the empirical data distribution.

## 3.1 NVDP Architecture

The architecture of our proposed NVDP model is illustrated in Figure 1. NVDP builds on top of a pretrained transformer encoder[2], which generates multi-vector transformer embeddings. These embeddings are processed by a single layer Transformer that incorporates our modified NVIB mechanism, projecting the embeddings into a mean, variance and pseudo-count for each input vector, which together determine the posterior distribution over sets of weighted vectors. Two key modifications are made to ensure privacy. First, we sample from this posterior distribution during both training and testing to generate a noisy, sanitized version of the embeddings. This stochastic bottleneck is the core of our privacy mechanism. Second, we process these noisy embeddings with a

---

[2]We use BERT-base-uncased model to generate embeddings.

Denoising Multi-Head Attention (MHA), but we remove the standard residual skip connection that would typically wrap this block. This architectural change is critical, as it prevents any un-sanitized information from the original embedding $x$ from leaking past the noisy latent representation and into the final output. This ensures that all shared information is passed exclusively through the privacy-preserving bottleneck.

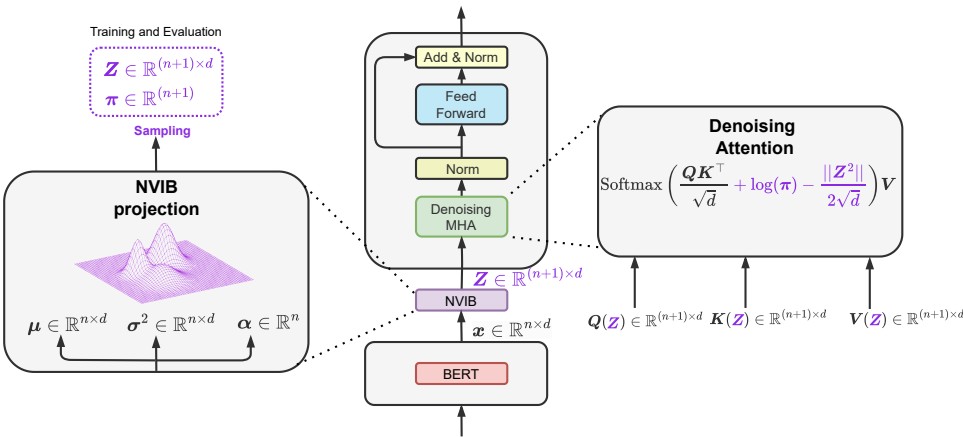

Figure 1: An NVDP model. The model projects an input embedding into the parameters of a posterior distribution. A key modification for privacy is that we sample from this latent distribution during both training and testing. The resulting noisy representation is processed by a Denoising MHA layer. To enforce the privacy bottleneck, the standard residual skip connection around the MHA is removed, preventing any un-sanitized information from bypassing the bottleneck.

More precisely, we first map the embedding from BERT (denoted as $\boldsymbol{x} \in \mathbb{R}^{n \times d}$ in Figure 1) into the parameters $(\boldsymbol{\alpha^q}, \boldsymbol{\mu^q}, (\boldsymbol{\sigma^q})^2)$ of a Dirichlet Process $DP(\alpha_0^q, G_0^q)$, as described in Section 2.2. Rather than interpreting these parameters as specifying an exact Dirichlet Process, we interpret them as specifying a sampling procedure which stochastically generates a finite sequence of weighted vectors $\boldsymbol{S} == (\boldsymbol{\pi}, \boldsymbol{Z})$, with probability $Q(\boldsymbol{S})$. Then we run this procedure to sample one such $\boldsymbol{S}$ and share $\boldsymbol{S}$. So we can think of the posterior distribution $Q(\boldsymbol{S})$ as the embedding of the text, and think of the sample $\boldsymbol{S}$ as a noisy version of that embedding.

### 3.2 MEASURING PRIVACY

In our local differential privacy setting, we want to make sure that any sample $\boldsymbol{S}$ from the embedding $Q(\boldsymbol{S})$ of a text $x$, could also have come from a different embedding $Q'(\boldsymbol{S})$ of some different text $\boldsymbol{x}'$. We measure this basic privacy criteria using the Rényi Divergence $D_\lambda(Q||Q')$, defined in Definition 2.1, to quantify the difference between these two sampling distributions. In this work, we measure privacy from two complementary perspectives, which differ in how we aggregate across the possible alternative inputs $\boldsymbol{x}'$. For both these measures, we report the worst case across the given input $\boldsymbol{x}$ (the average case is reported in Table 2 in Appendix A).

The first approach measures the worst case also across all alternative inputs $\boldsymbol{x}'$, as in standard RDP. We do not assume any specific notion of adjacency between examples. In our experiments, we report the maximum Renyi divergence over all input pairs as the **RDP** measure (the average case is reported in Table 2 in Appendix A ).

The second approach measures an aggregation across all alternative inputs $\boldsymbol{x}'$, as is done in BDP. This measures how much an individual's output $Q$ stands out from the crowd, thereby measuring the risk of de-anonymization. As discussed in Section 2.1, we aggregate the RD values (as calculated for the first measure) across examples $\boldsymbol{x}'$ and convert to an interpretable $(\epsilon_\mu, \delta_\mu)$-style privacy loss for the **BDP** measure.

To calculate the Rényi divergence (RD) between the posterior embedding $Q(\boldsymbol{S})$ for input $x$ and an alternative embedding $Q'(\boldsymbol{S})$ for input $\boldsymbol{x}'$, we first specify a sampling procedure which approximates sampling from a Dirichlet Process, and then calculate the RD for that sampling procedure applied to

$Q$ and $Q'$. This is an upper bound on the RD between the two Dirichlet Processes, since a sequence of weighted vectors is more specific than a mixture distribution, but it is appropriate for our privacy measure because it is this sequence which is actually shared. So the RD between sampling two different finite sequences of weighted vectors from two different inputs is an accurate assessment of how easily an observer can distinguish between samples from one text's embedding and samples from the other text's embedding, providing a measure of privacy.

### 3.3 RÉNYI DIVERGENCE BETWEEN TWO SAMPLING DISTRIBUTIONS

Since the theory of NVIB assumes that the posterior distribution which embeds the information about the input data is a DP, $Q = DP(\alpha_0^q, G_0^q)$, we want a sampling procedure which generates finite sequences of weighted vectors which approximate sampling from this DP. For $DP(\alpha_0^q, G_0^q)$, defined in equation 4, weights can be sampled using a stick breaking process parameterised by $\alpha_0^q$, and vectors can be sampled independently by first sampling a component $i$ from the mixture distribution $G_0^q$ and then sampling a vector from that component's distribution $G_i^q = \mathcal{N}(\mu_i^q, (\sigma_i^q)^2)$. But sampling from the discrete list of components $i$ makes it difficult for NVIB to do backprop through the sampling step, so instead Henderson & Fehr (2023) propose a sampling method which first samples the total weight assigned to all the vectors from a given component $i$, and then samples weighted vectors independently from the component's distribution $G_i^q$. They prove that the DP distribution in equation 4 is equivalent to such a facorised distribution as follows:

$$F \;=\; \sum_{i=1}^{n+1} \rho_i F_i \qquad \boldsymbol{\rho} \;\sim\; \mathrm{Dir}(\alpha_1^q, \ldots, \alpha_{n+1}^q) \qquad F_i \;=\; DP(G_i^q, \alpha_i^q) \qquad (6)$$

In particular, Henderson & Fehr (2023) show that sampling the total weights $\boldsymbol{\rho_i}$ across $i$ can be done by sampling from a Dirichlet distribution parameterised by $\boldsymbol{\alpha^q}$, and the distribution of weighted vectors sampled from each component is another Dirichlet process. To generate a sample with a finite number of vectors, they assume a given bound $\kappa_i$ on the number of vectors sampled from each component $G_i^q$, and sample their weights with a symetric Dirichlet distribution $\mathrm{Dir}(\frac{\alpha_i^q}{\kappa_i}, \overset{\kappa_i}{\ldots}, \frac{\alpha_i^q}{\kappa_i})$. In our experiments, one vector is sampled from each component, so $\kappa_i = 1$.

The different weighted vectors sampled from a Dirichlet Process specify a mixture distribution, which is permutation invariant. However, when a sample is generated with our sampling procedure, the weighted vectors are output in a specific order. We could have a better approximation to a Dirichlet Process by randomly permuting the weighted vectors, but for simplicity we output them in the order of the tokens in the sentence. This simplifies the computation of the RD between two samples because it implies that sampled vectors for two different inputs can be aligned by their token position, which gives us an upper bound on the Dirichlet Process case, since the ordered list is more informative.[3] This gives us the following formula for calculating the Rényi divergence between samples from two different input embeddings.

$$D_\lambda(\mathrm{DP}(G_0^q, \alpha_0^q) \,\|\, \mathrm{DP}(G_0^{q'}, \alpha_0^{q'})) \tag{7}$$

$$\leq - \left( \frac{1}{\lambda-1} \log \Gamma\left(\lambda \alpha_0^q - (\lambda-1)\alpha_0^{q'}\right) \;+\; \log \Gamma(\alpha_0^{q'}) - \frac{\lambda}{\lambda-1} \log \Gamma(\alpha_0^q) \right)$$

$$+ \sum_{i=1}^{n+1} \kappa_i \left( \frac{1}{\lambda-1} \log \Gamma(\lambda \frac{\alpha_i^q}{\kappa_i} - (\lambda-1)\frac{\alpha_i^{q'}}{\kappa_i}) \;+\; \log \Gamma(\frac{\alpha_i^{q'}}{\kappa_i}) - \frac{\lambda}{\lambda-1} \log \Gamma(\frac{\alpha_i^q}{\kappa_i}) \right)$$

$$+ \sum_{i=1}^{n+1} \kappa_i \left( \frac{\lambda}{2} \left\| \frac{\boldsymbol{\mu_i^q} - \boldsymbol{\mu_i^{q'}}}{\boldsymbol{\sigma_i'}} \right\|^2 \;+\; \frac{1}{1-\lambda} \mathbf{1} \left( \log \frac{\boldsymbol{\sigma_i'}}{(\boldsymbol{\sigma_0^p})^{(1-\lambda)}(\boldsymbol{\sigma_i^q})^\lambda} \right) \right)$$

where $\boldsymbol{\sigma_i'} = \sqrt{(1-\lambda)(\boldsymbol{\sigma_i^{q'}})^2 + \lambda(\boldsymbol{\sigma_i^q})^2}$, and $\mathbf{1}$ is a vector of 1s.

---

[3]To handle the case where the two examples have different numbers of tokens, we pad the inputs sequences so that they all have the same length, and then assume that pad tokens have parameters $\mu_i = \mathbf{0}$, $\sigma_i = \mathbf{1}$, $\alpha_i = \mathbf{0}$. Alternatively, we could have defined adjacent inputs as all having the same length, but this approach is more appropriate for the BDP measure and still gives a meaningful privacy measure. We leave better bounds on the RD between samples from Dirichlet Processes to future work.

# 4 EXPERIMENTS

We empirically evaluate NVDP on NLP tasks from the GLUE benchmark (Wang et al., 2018) in terms of both accuracy and privacy.

**Datasets** We evaluate the performance of NVDP on the General Language Understanding Evaluation (GLUE) (Wang et al., 2018) benchmark, which is a collection of different natural language understanding tasks including similarity and paraphrasing tasks, text classification, and natural language inference (NLI). For NLI, we experiment on QNLI (Rajpurkar et al., 2016) and RTE (Dagan et al., 2005). For paraphrase detection, we evaluate on MRPC (Dolan & Brockett, 2005), STS-B (Cer et al., 2017), and QQP (Chen et al., 2018). For text classification, we evaluate on SST-2 (Socher et al., 2013).

**Base Model** We use BERTBase, which is 12 layers and 110M parameters. We use the following hyper-parameters for fine-tuning BERT: a sequence length of $512$, a train batch size of $64$ and evaluation batch size of $8$. We use the stable variant of the Adam optimizer (Zhang et al., 2020; Mosbach et al., 2020) with the learning rate of $2e-7$ through all experiments. We use a warm-up step of $0.2$.

**Baselines** We compare our method with the prior state-of-the-art baseline and previous regularization techniques:

- Baseline: we use vanilla BERTBase model without any regularizaton technique.

- Regularization: we use Dropout (Srivastava et al., 2014)& Weight Decay (WD) (Krogh & Hertz, 1991). Drouput is a stochastic regularization technique widely used in various large-scale language models to reduce overfitting (Devlin et al., 2019; Yang, 2019; Ashish, 2017). A dropout of 0.1 is applied across all layers of BERT. Another technique is weight decay (WD), which helps improve generalization by penalizing large weights with a term $\frac{\lambda}{2}||\boldsymbol{y}||$ in the loss function, where $\lambda$ controls the regularization strength. For fine-tuning pretrained models, a modified WD to use $\lambda||\boldsymbol{y} - \boldsymbol{y}0||$ is used where $\boldsymbol{y}0$ represents the pretrained weights (Chelba & Acero, 2004; Daumé III, 2007). Lee et al. (2019) showed that this adjusted version of WD enhances fine-tuning for BERT, particularly on smaller datasets, compared to the traditional approach. A WD of $0.01$ is applied.

**Ablation** We consider a simplified version of our proposed model where the noise introduced with NVIB is replaced with noise introduced by vector-space VIB applied to each token vector independently, which we call the variational transformer differential privacy (VTDP) model. This provides a strong reference model which is the same as our NVDP model but without the nonparametric regularisation provided by NVIB.

- VTDP: We utilize BERTBase with an additional Variational Information Bottleneck (VIB) layer to implement the information bottleneck principle outlined by Mahabadi et al. (2021). The VIB layer encodes input representations in a compressed latent space, leveraging VIB to learn a stochastic latent variable. This variable captures task-relevant information while filtering out irrelevant features, enhancing generalization. This integration enables effective fine-tuning, particularly in low-resource scenarios, by prioritizing essential information and mitigating overfitting risks. The compressed latent representation is compared to a Gaussian prior, ensuring a structured and regularized latent space. The Rényi Differential Privacy (RDP) guarantee, defined between Gaussian distributions, follows the formulation in Equation 10 of Van Erven & Harremos (2014):

$$D_\lambda(\mathcal{N}(\boldsymbol{\mu}_i^q, \boldsymbol{\sigma}_i^q)||\mathcal{N}(\boldsymbol{\mu}_0^p, \boldsymbol{\sigma}_0^p)) \tag{8}$$

$$= \frac{\lambda}{2}\left\|\frac{\boldsymbol{\mu}_i^q - \boldsymbol{\mu}_0^p}{\boldsymbol{\sigma}_i'}\right\|^2 + \frac{1}{1 - \lambda}\mathbf{1}\left(\log\frac{\boldsymbol{\sigma}_i'}{(\boldsymbol{\sigma}_0^p)^{(1-\lambda)}(\boldsymbol{\sigma}_i^q)^\lambda}\right)$$

Table 1: Privacy-utility trade-off on GLUE tasks for BERT-Base. We compare our proposed **NVDP** model against non-private baselines and a VIB-based ablation (**VTDP**). For each private model, we report its best-achieved utility score alongside privacy guarantees. Privacy is measured via Bayesian Differential Privacy (BDP ↓) and Rényi Divergence (RD ↓). Lower privacy values are better. The best-performing private model per task is **bolded**.

| Dataset | Metric | Baselines (Non-Private) | | VTDP (Ablation) | | | NVDP (Ours) | | |
|---|---|---|---|---|---|---|---|---|---|
| | | Base | +REG | Score | BDP↓ | RD (max) ↓ | Score | BDP↓ | RD (max) ↓ |
| MRPC | Accuracy | 81.2 | 82.4 | 81.1 | 11.50 | 1.20 | **83.0** | **10.70** | **0.34** |
| | F1 Score | 86.0 | 87.6 | 86.5 | 11.50 | 1.20 | **87.5** | **10.70** | **0.34** |
| STS-B | Pearson | 86.0 | 85.7 | 83.6 | 22.20 | 6.61 | **85.2** | **20.93** | **1.41** |
| | Spearman | 84.9 | 84.5 | 82.3 | 22.20 | 6.61 | **84.0** | **20.93** | **1.41** |
| RTE | Accuracy | 65.9 | 66.3 | 64.1 | 11.50 | 1.94 | **64.8** | **10.90** | **1.66** |
| QQP | Accuracy | 87.8 | 88.4 | 87.6 | 15.52 | 0.85 | **88.3** | **13.01** | **1.14** |
| | F1 Score | 68.4 | 69.4 | 67.4 | 15.52 | 0.85 | **68.9** | **13.01** | **1.14** |
| QNLI | Accuracy | 89.0 | 89.7 | 87.1 | 16.90 | 1.80 | **89.5** | **12.10** | **0.75** |
| SST-2 | Accuracy | 92.9 | 91.9 | **92.3** | 10.90 | 0.37 | 91.7 | **10.90** | **0.19** |

### 4.1 Results on the GLUE Benchmark

Table 1 presents a summary of the best privacy-utility trade-off achieved by each private model, while Figure 2 visualize the full trade-off curve using the BDP measure.

**Experimental Protocol.** For each model, we perform five independent runs and select the best-performing run on the validation set for final evaluation on the test set. The NVDP and VTDP models provide privacy by applying the same learned stochastic mapping at test time, ensuring that only noisy, sanitized embeddings are shared. The utility of these private embeddings is measured by evaluating the classifier learned during training. To quantify the privacy loss shown in Table 1, we fix the Rényi order to $\lambda = 1.1$ (in equation 7) and the BDP failure probability to $\delta_\mu = 10^{-5}$ (in equation 3) and report the worst-case divergence across all test set pairs. Full results for all regularization strengths are provided in Appendix A.

**Utility and Regularization Analysis.** The results in Table 1 show that the NVDP model also functions as an effective regularization technique. The utility of the NVDP model is highly competitive with the non-private, regularized baseline (+REG). On some tasks, such as MRPC, NVDP achieves a higher accuracy (83.0% vs. 82.4%), while on others, like STS-B, it performs comparably (85.2 vs. 85.7). In stark contrast, the VTDP ablation generally struggles to maintain the same level of utility. For example, on MRPC, the best VTDP accuracy is only 81.1%, a significant drop from both the +REG baseline and our NVDP model. The only exception is SST-2, where VTDP achieves a slightly higher accuracy, but as discussed below, this comes at a significant cost in terms of the underlying privacy divergence. This demonstrates that NVDP is better able to preserve task-critical information while regularizing the model.

### 4.2 Privacy-Utility Trade-off Analysis

We analyze the privacy-utility trade-offs of our models presented in table 1. For the Bayesian Differential Privacy guarantee, the results show a clear advantage for our proposed model. Figure 2 plots accuracy against the BDP measure (detailed results are given in Table 2 in Appendix A). Across all datasets, the NVDP models (blue curves) consistently occupy the most favorable region of the plot—closest to the top-right corner—indicating a superior trade-off compared to the VTDP ablation (red curves). For instance, on MRPC, NVDP reaches 83% accuracy with an BDP($\epsilon_\mu$) of 10.7. To achieve a comparable privacy budget (10.6), the VTDP model's accuracy drops to just 74.8%, underscoring the superior privacy-utility frontier of our NVDP method. This suggests that NVIB's

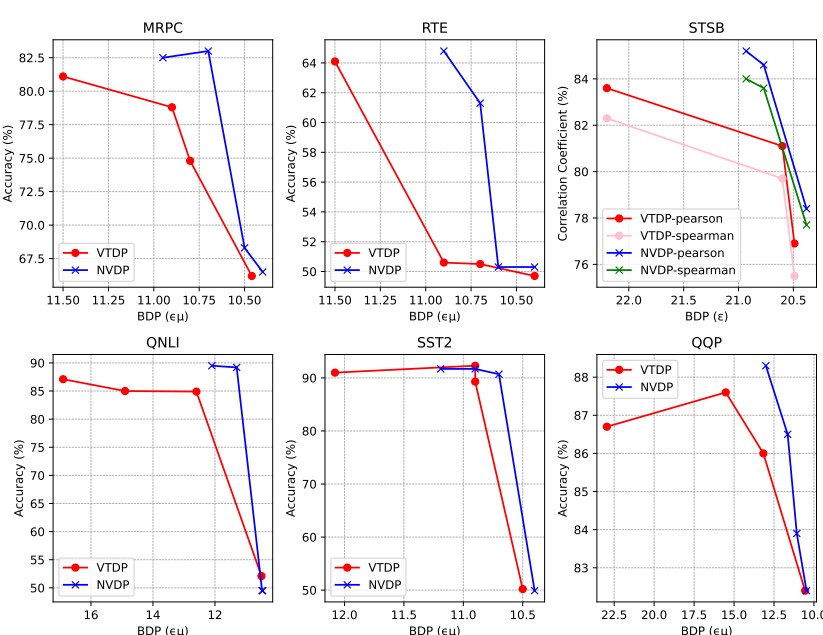

Figure 2: Accuracy versus Bayesian Differential Privacy ($\epsilon_\mu$). The BDP budget ($\epsilon_\mu$) is calculated by finding the tightest privacy guarantee for a fixed $\delta_\mu = 1e - 5$. This provides a more interpretable view of the privacy-utility trade-off, where lower ($\epsilon_\mu$) values signify stronger, more practical privacy guarantees. The NVDP model consistently achieves better privacy-utility points than the VTDP ablation.

mechanism is more effective at removing extraneous, potentially privacy-sensitive information while retaining utility.

This conclusion is further strengthened by the Rényi Divergence (RD) values reported in Table 1 (RD curves are plotted in Figure 3 in Appendix A). As a direct measure of distinguishability, the RD reveals an even larger performance gap between the models. In the same MRPC comparison, NVDP's worst-case RD is 0.34, whereas VTDP's is 1.20, indicating substantially lower raw information leakage. The benefit is particularly clear on SST-2; while both models can be tuned to an identical BDP of 10.90, the underlying RD for our NVDP model is nearly half that of the VTDP ablation (0.19 vs. 0.37). Together, these results confirm that NVIB's mechanism is more effective at removing privacy-sensitive information while retaining utility.

## 5 CONCLUSION

We propose a model that addresses the privacy concerns in sharing data for deep learning by integrating a nonparametric variational information bottleneck layer into the transformer architecture and sharing transformer embeddings. By leveraging NVIB and differential privacy, measured by Rényi divergence, our model, nonparametric variational differential privacy, is able to share embeddings of data while providing strong privacy guarantees and maintaining downstream model utility.

Our experimental results demonstrate the effectiveness of this approach by evaluating privacy. We confirmed that NVDP consistently controls information leakage more effectively than a strong VIB-based ablation across a range of GLUE tasks. Crucially, by converting our Rényi Divergence measurements into interpretable ($\epsilon_\mu$, $\lambda_\mu$)-Bayesian Differential Privacy guarantees, we have shown that our model can achieve strong, practical privacy budgets while maintaining high model utility. This is a significant step towards deploying privacy-preserving transformer embeddings in real-world applications where clear and meaningful privacy assurances are required.

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

# A ADDITIONAL PRIVACY ANALYSIS

This appendix provides supplementary results to complement the analysis in the main text. Table 2 presents the comprehensive numerical results presented in Table 1, detailing the utility and privacy metrics for all tested KL-divergence regularization weights. So, this table provide the complete data used to generate the summary results in Table 1. Figure 3 visualize the privacy-utility trade-offs for the worst case *max* Rényi Divergence. These results demonstrate that our NVDP model consistently outperforms the VTDP ablation under all evaluation settings.

Table 2: Full experimental results for BERT-Base on GLUE, showing the complete privacy-utility trade-off across all tested KL-divergence regularization weights ($\lambda_G, \lambda_D$). For each model, we report Utility (task-specific scores) and privacy metrics BDP ($\epsilon_\mu$) and RD. The RD metric are reported as max/avg, representing the worst-case and average-case divergence across all example pairs, respectively. Lower values are better for all privacy metrics ($\downarrow$).

| Model | Metric | MRPC | STS-B | RTE | QQP | QNLI | SST-2 |
|---|---|---|---|---|---|---|---|
| | | Acc / F1 | Pearson / Spearman | Accuracy | Acc / F1 | Accuracy | Accuracy |
| BERT$_{\text{Base}}$ | Utility | 81.2 / 86.0 | 86.0 / 84.9 | 65.9 | 87.8 / 68.4 | 89.0 | 92.9 |
| | Privacy | - | - | - | - | - | - |
| +REG | Utility | 82.4 / 87.6 | 85.7 / 84.5 | 66.3 | 88.4 / 69.4 | 89.7 | 91.9 |
| | Privacy | - | - | - | - | - | - |
| NVDP$_{\lambda=1e-3}$ | Utility | 82.5 / 87.1 | 85.2 / 84.0 | 64.8 | 88.3 / 68.9 | 89.5 | 91.7 |
| | BDP ($\epsilon_\mu$) $\downarrow$ | 10.95 | 20.93 | 10.90 | 13.01 | 12.10 | 11.19 |
| | RD (max/avg) $\downarrow$ | 0.89 / 0.06 | 1.41 / 0.10 | 1.66 / 0.12 | 1.143 / 0.031 | 0.75 / 0.06 | 1.00 / 0.06 |
| NVDP$_{\lambda=1e-2}$ | Utility | 83.0 / 87.5 | 84.6 / 83.6 | 61.3 | 86.5 / 67.9 | 89.2 | 91.7 |
| | BDP ($\epsilon_\mu$) $\downarrow$ | 10.70 | 20.77 | 10.70 | 11.64 | 11.30 | 10.90 |
| | RD (max/avg) $\downarrow$ | 0.34 / 0.02 | 1.22 / 0.05 | 0.87 / 0.04 | 0.53 / 0.028 | 0.71 / 0.09 | 0.19 / 0.01 |
| NVDP$_{\lambda=1e-1}$ | Utility | 68.3 / 80.2 | 78.4 / 77.7 | 50.3 | 83.9 / 65.5 | 49.5 | 90.7 |
| | BDP ($\epsilon_\mu$) $\downarrow$ | 10.50 | 20.38 | 10.60 | 11.08 | 10.48 | 10.70 |
| | RD (max/avg) $\downarrow$ | 0.04 / 0.01 | 0.22 / 0.01 | 0.10 / 0.01 | 0.37 / 0.02 | 0.016 / 0.003 | 0.016 / 0.004 |
| NVDP$_{\lambda=1}$ | Utility | 66.5 / 79.9 | 82.7 / 82.9 | 50.3 | 82.4 / 0 | 49.5 | 49.9 |
| | BDP ($\epsilon_\mu$) $\downarrow$ | 10.40 | 18.65 | 10.40 | 10.46 | 10.46 | 10.40 |
| | RD (max/avg) $\downarrow$ | 0.008 / 0.002 | 0.03 / 0.004 | 0.005 / 0 | 0.006 / 0.001 | 0.007 / 0.001 | 0.01 / 0.002 |
| VTDP$_{\lambda=1e-3}$ | Utility | 81.1 / 86.5 | 83.6 / 82.3 | 64.1 | 86.7 / 67.6 | 87.1 | 91.0 |
| | BDP ($\epsilon_\mu$) $\downarrow$ | 11.50 | 22.20 | 11.50 | 22.95 | 16.90 | 12.08 |
| | RD (max/avg) $\downarrow$ | 1.2 / 0.37 | 6.61 / 0.77 | 1.94 / 0.44 | 3.33 / 0.59 | 1.80 / 0.68 | 1.45 / 0.54 |
| VTDP$_{\lambda=1e-2}$ | Utility | 78.8 / 84.9 | 81.1 / 79.7 | 50.6 | 87.6 / 67.4 | 85.0 | 92.3 |
| | BDP ($\epsilon_\mu$) $\downarrow$ | 10.90 | 20.60 | 10.90 | 10.90 | 14.90 | 10.90 |
| | RD (max/avg) $\downarrow$ | 0.43 / 0.12 | 1.33 / 0.13 | 1.62 / 0.33 | 0.85 / 0.14 | 0.39 / 0.14 | 0.37 / 0.15 |
| VTDP$_{\lambda=1e-1}$ | Utility | 74.8 / 83.3 | 76.9 / 75.5 | 50.5 | 86.0 / 62.3 | 84.9 | 89.3 |
| | BDP ($\epsilon_\mu$) $\downarrow$ | 10.60 | 20.49 | 10.70 | 13.16 | 12.60 | 10.90 |
| | RD (max/avg) $\downarrow$ | 0.049 / 0.018 | 0.33 / 0.04 | 0.18 / 0.04 | 0.09 / 0.019 | 0.04 / 0.019 | 0.13 / 0.05 |
| VTDP$_{\lambda=1}$ | Utility | 66.2 / 79.6 | 52.6 / 51.6 | 49.7 | 82.4 / 0.3 | 52.1 | 50.2 |
| | BDP ($\epsilon_\mu$) $\downarrow$ | 10.46 | 20.30 | 10.40 | 10.55 | 10.50 | 10.50 |
| | RD (max/avg) $\downarrow$ | 0 / 0 | 0.038 / 0.005 | 0 / 0 | 0 / 0 | 0 / 0 | 0 / 0 |

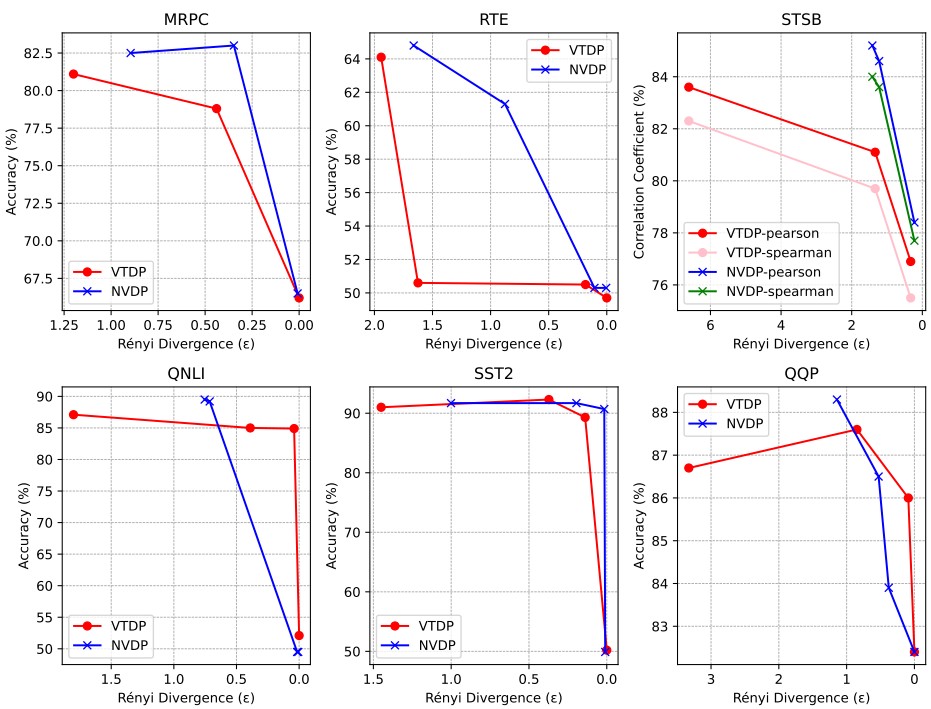

Figure 3: Accuracy versus **maximum** Rényi Divergence (RD) illustrating the worst-case privacy-utility tradeoff. Each point corresponds to a different KL regularization weight, where stronger regularization leads to better privacy (lower RD). The most favorable models are those closest to the upper-right corner.

