# OpenReview forum: "Differential Privacy for Transformer Embeddings  with Nonparametric Variational Information Bottleneck"
_ICLR.cc/2026/Conference — Submitted to ICLR 2026_

### Official Review · Reviewer_Q2oc · 2025-10-25

**Soundness:** 2
**Presentation:** 2
**Contribution:** 2
**Rating:** 2
**Confidence:** 4

**Summary:**

This paper proposes NVDP (Nonparametric Variational Differential Privacy), a new method to provide local differential privacy for transformer embeddings. The motivation stems from the observation that transformer hidden states may leak sensitive information, allowing adversaries to reconstruct or infer private attributes. To mitigate this, the authors integrate a Nonparametric Variational Information Bottleneck into a transformer encoder. NVIB, based on a Dirichlet Process latent prior, stochastically samples weighted vectors (token-level embeddings) that preserve utility while controlling information flow. The method injects learned, task-calibrated noise into embeddings, and privacy is quantified via Rényi Differential Privacy and converted into interpretable Bayesian Differential Privacy guarantees.

**Strengths:**

- The paper is well structured, with clear separation between background, method, and experiments.
- Introduces the first integration of NVIB and differential privacy within transformer architectures, bridging information bottleneck theory and formal privacy guarantees.
- Provides a viable way to share transformer embeddings safely.

**Weaknesses:**

- NVIB sampling and RD computation over all input pairs are expensive (O(n²) pairs). No runtime, memory, or scalability analysis is provided.
- While the paper repeatedly claims local DP, its formulation (sampling embeddings within the model) seems effectively performs mechanism-level DP, not user-level DP.

**Questions:**

- How tight is your analytical RD bound (Eq. 7)? Any empirical validation or confidence intervals?
- How does NVDP compare with other DP mechanisms on the same tasks?
- What is the computational overhead (training time per epoch, GPU hours)?

---

> ### Author Response · Authors · 2025-11-24
>
> Thank you for your clear precise review.  We reply to the three questions, thereby also addressing the two weaknesses.
>
>
> Q: How tight is your analytical RD bound (Eq. 7)? Any empirical validation or confidence intervals?
>
>  The analytical bound in Equation (7) provides a formal upper bound on the Rényi Divergence for our specific sampling mechanism. While a theoretical analysis of the bound's tightness is a complex direction for future work, its practical utility is demonstrated empirically throughout our experiments. The bound proves to be a sensitive and consistent metric, as it clearly distinguishes between models with different privacy levels and impacts on model utility.
>
> Q: How does NVDP compare with other DP mechanisms on the same tasks?
>
> We compare NVDP primarily against our VTDP ablation, a strong baseline that adds calibrated Gaussian noise to embeddings, similar in principle to other recent methods. Our results consistently show that NVDP achieves a superior privacy-utility trade-off. We will run other baselines, such as using single-vector embeddings, but we expect very similar results to VTDP.
>
>
>
> Q: What is the computational overhead (training time per epoch, GPU hours)?
>
> The overhead is minimal. On low-resource datasets (containing 2–4k training examples), training time is approximately 5-10 minutes per 10 epochs on a single GPU, confirming the efficiency of our method.

---

### Official Review · Reviewer_kjrT · 2025-10-31

**Soundness:** 3
**Presentation:** 3
**Contribution:** 2
**Rating:** 4
**Confidence:** 3

**Summary:**

This paper proposes NVDP (Nonparametric Variational Differential Privacy), a framework that integrates Bayesian DP and RDP with a nonparametric variational information bottleneck (VIB). The approach aims to achieve privacy-preserving embeddings by learning stochastic representations that limit mutual information between inputs and embeddings. NVDP introduces internal probabilistic noise through a variational layer parameterized by $\mu, \sigma, \alpha$. The model thus learns to generate privacy-compliant embeddings by minimizing the expected RDP distance between output distributions across data samples.

**Strengths:**

- S1: Conceptually appealing integration: The unification of RDP and BDP within an information bottleneck formulation is a novel and elegant conceptual contribution. It provides a fresh probabilistic perspective on DP, reframing privacy as information compression rather than explicit noise injection.
- S2: Learning-based noise adaptation: Unlike conventional DP mechanisms with fixed noise levels, NVDP allows the model to adapt its internal noise dynamically via learned parameters $(\sigma(x), \alpha(x))$. This could, in principle, improve the privacy–utility trade-off.

**Weaknesses:**

- W1: Lack of practical validation: The experiments do not convincingly demonstrate real-world usefulness. It remains unclear for which downstream applications (e.g., classification, retrieval, or fine-tuning) the learned embeddings preserve performance while providing privacy.
- W2: While $\epsilon$ is computed through RDP/BDP metrics, it is not specified or controlled in the same way as standard DP. The reader cannot interpret what an obtained $\epsilon$ actually means in practical or regulatory terms.
- W3: Since the model operates under a local DP assumption, each client may achieve a different effective privacy strength. The implications for consistency, fairness, and aggregation are not discussed.
- W4: Approximate, not formal DP guarantees: The framework measures privacy using divergence bounds but does not prove formal composition or post-processing guarantees. As a result, it is more accurately described as ``DP-inspired” rather than strictly DP-compliant.
- W5: Despite the emphasis on ``Transformer embeddings,” the proposed method does not exploit any specific properties of Transformer architectures. The model merely applies the NVDP layer on top of general embeddings, meaning that the approach could equally apply to CNNs or MLP-based encoders.

**Questions:**

Address W1-5.

---

> ### Author Response · Authors · 2025-11-24
>
> Thank you for your thoughtful review.  We reply to weaknesses W1 - W5 as specified in the review.
>
> W1: Our experiments are, in fact, designed to directly measure the real-world usefulness for the primary application of classification. The GLUE benchmark, on which all our results are based, is a standard and widely-used suite of diverse text classification tasks. Our evaluation protocol directly addresses the reviewer's concern: (1) We take a model pre-trained for a specific GLUE classification task. (2) We process the test set data through our NVDP layer to generate private embeddings. (3) We then evaluate the performance of the original, unmodified classifier on these private embeddings. The accuracy scores reported in Table 1 therefore represent the preserved performance on downstream classification tasks.
>
>
> W2: It is correct that we do not directly "control" the final $\epsilon_\mu$ as a fixed input. Instead, our method uses the NVIB hyperparameters $(\lambda_D, \lambda_G)$ to tune the privacy-utility trade-off, and the resulting $\epsilon_\mu$ is a measurement of the privacy loss for a given trained model, a standard practice in DP-ML. Regarding interpretation, the $\epsilon$ values from RDP are directly comparable to standard DP epsilon. Notably, recent works in the same domain, such as Meehan et al. (2022) [2], demonstrate that $\epsilon$ values <10 are considered meaningful for complex NLP tasks. Our RDP results, which fall within this range, should therefore be interpreted as providing strong, practical privacy guarantees according to current community standards.
>
>
> W3: To address the crucial issue of providing a consistent guarantee that holds for every user, our entire privacy analysis is based on the worst-case privacy loss. The "max" RD values reported in Table 1 represent the maximum observed divergence over all pairs in the test set, and the BDP budget $\epsilon\mu$ is calculated from this worst-case value. This ensures that our reported $(\epsilon\mu, \delta_\mu)$-guarantee is a valid LDP guarantee that formally protects every individual, including the most vulnerable outliers. We also report the average RD to provide a more complete picture of the typical performance. While our worst-case guarantee protects all users, the fact that "inliers" may be receiving unnecessarily strong privacy while "outliers" receive just the minimum is an important fairness consideration. Analyzing and potentially mitigating this variance is a research topic in its own. Similarly, the formal composition of data with heterogeneous privacy levels is a complex theoretical question. We agree that these are important limitations that we should discuss.
>
>
> W4: We respectfully disagree with the characterization of our guarantees as "DP-inspired" rather than formal. Our privacy analysis is grounded in the formal frameworks of Rényi Differential Privacy (RDP) and Bayesian Differential Privacy (BDP). The theoretical formulas for BDP are directly provided in the BDP paper and we can provide the ones for RDP.  Our method inherits all the formal properties that have been shown in previous work on this privacy framework, including formal composition or post-processing guarantees.
>
>
> W5: We use "Transformer embeddings" as an intuitive way to refer to the multi-vector embeddings output by a Transformer encoder, which can be accessed with attention.  They might better be referred to as "attention-based embeddings".  We apologise if this was not clear.  Indeed, one of the main novelties of our method is precisely that it applies to these multi-vector embeddings, rather than the single-vector sentence embeddings used in much previous work.  This makes our NVDP layer particularly appropriate for privatising the output embeddings of one Transformer for input via attention to another Transformer.  But we agree that our method is a general, modular approach for privatizing any embedding which is a sequence of vectors. As such, it is indeed applicable to the token-level outputs of other modern sequential encoders, such as LSTMs or CNNs. However, the reviewer suggests that it could apply to "MLP-based encoders" which is not the case, as a standard MLP produces a single fixed-size vector for the entire input, whereas our NVDP layer is specifically designed to operate on a sequence of vectors (n x d tensor), which is the standard output format for deep sequence models like Transformers.

---

### Official Review · Reviewer_6gyN · 2025-11-01

**Soundness:** 3
**Presentation:** 2
**Contribution:** 2
**Rating:** 4
**Confidence:** 3

**Summary:**

The paper proposes Nonparametric Variational Differential Privacy (NVDP) that offers privacy protection on transformer embeddings. It modifies NVIB by noisy embedding sampling and subsequent denoising, providing RDP and BDP guarantee. Experiment results demonstrate that NVDP effectively balance privacy and utility.

**Strengths:**

- The paper tackles a real-world problem of privacy-preserving data sharing.
- This paper adapts NVIB to provide formal privacy guarantee during embedding sharing.
- NVDP offers better privacy and utility trade-off compared with the baselines.

**Weaknesses:**

- The experiment lacks comparison with existing DP baselines [1][2].
- The discussion on neighboring dataset is vague. It is unclear how the neighbors map to the token/sentence scenario.
- The experiment lacks attack analysis on the privatized embeddings, such as

[1] Du, M., Yue, X., Chow, S. S., & Sun, H. (2023, April). Sanitizing sentence embeddings (and labels) for local differential privacy. In Proceedings of the ACM Web Conference 2023 (pp. 2349-2359).

[2] Meehan, C., Mrini, K., & Chaudhuri, K. (2022, May). Sentence-level Privacy for Document Embeddings. In Proceedings of the 60th Annual Meeting of the Association for Computational Linguistics (Volume 1: Long Papers) (pp. 3367-3380).

**Questions:**

NA

---

> ### Author Response · Authors · 2025-11-24
>
> We thank the reviewer for their clear precise review.  We reply to the three weaknesses.
>
> Weakness: The experiment lacks comparison with existing DP baselines [1],[2].
>
> Thank you for the useful references. We note that both [1] and [2] operate in the same setting as our work (LDP for text embeddings), confirming the relevance of our problem space.  As suggested, we are running additional baselines which provide a controlled comparison with methods like [1] and [2].  But our VTDP ablation serves as a strong VIB-based Gaussian noise baseline, conceptually similar to methods in [1] and [2]. Our results consistently show that our adaptive NVDP method achieves a superior privacy-utility trade-off compared to this strong baseline.
> Our method is distinguished from [1] and [2] in two primary novelties: our attention-based text embeddings (i.e.\ multiple vectors), and our task-aware calibration of randomness, both made possible by the Nonparametric Variational Information Bottleneck (NVIB).  This differs from the single-vector embeddings and the task-agnostic noise mechanisms in the cited works, so such a comparison would be an interesting ablation.
>
>
> Weakness: The discussion on neighboring dataset is vague. It is unclear how the neighbors map to the token/sentence scenario.
>
> We agree that our discussion of the privacy model was not sufficiently precise. Our framework operates under the Local Differential Privacy (LDP) model, where the guarantee concerns the indistinguishability of the mechanism's output for any two inputs x and x’.  Thus, all inputs x are considered neighbors of all other inputs x'.
>
>
> Weakness: The experiment lacks attack analysis on the privatized embeddings, such as in [1] and [2].
>
> We agree that complementing our formal privacy guarantees with an empirical evaluation against practical privacy attacks (e.g., membership inference) would provide a more complete picture of our model's real-world security. However, our paper's primary contribution is the introduction and theoretical analysis of the NVDP mechanism itself. Given this focus, we considered a full-scale empirical attack analysis to be beyond the scope of this work.

---

### Meta-Review · Area_Chair_oMM9 · 2026-01-07

**Summary:**

This paper provides a machine learning approach to learn stochastic representations that are designed to have low mutual information with he input and hence private. Two reviewers had concerns about the level of formalism in the paper (no clarity about the mechanism-level or user-level privacy) and one questioned the validity of the bounds.

**Reviewer Concerns:**

- [Reviewer 6gyN] Comparison with baseline
- [Reviewer 6gyN, kjrT, Q2oc] Vague definition
- [Reviewer 6gyN, kjrT, Q2oc]  Practicality in real world applications

**Reviewer Scores:**

I believe reviewers would remain on their current scores after the rebuttal as I don't find authors rebuttal satisfactory.

---

### Decision · Program_Chairs · 2026-01-26

Reject